# Exploring Training Data Attribution under Limited Access Constraints

## Abstract

Training data attribution (TDA) plays a critical role in understanding the influence of individual training data points on model predictions. Gradient-based TDA methods, popularized by *influence function* for their superior performance, have been widely applied in data selection, data cleaning, data economics, and fact tracing. However, in real-world scenarios where commercial models are not publicly accessible and computational resources are limited, existing TDA methods are often constrained by their reliance on full model access and high computational costs. This poses significant challenges to the broader adoption of TDA in practical applications.

In this work, we present a systematic study of TDA methods under various access and resource constraints. We investigate the feasibility of performing TDA under varying levels of access constraints by leveraging appropriately designed solutions such as proxy models. Besides, we demonstrate that attribution scores obtained from models without prior training on the target dataset remain informative across a range of tasks, which is useful for scenarios where computational resources are limited. Our findings provide practical guidance for deploying TDA in real-world environments, aiming to improve feasibility and efficiency under limited access.

## 1 Introduction

Data-centric artificial intelligence (AI) is an emerging research paradigm that emphasizes systematically designing, curating, and improving data for AI systems (Zha et al., 2025). The pivotal role training data plays in shaping the performance and behavior of modern AI systems motivates the development of data-centric AI. Training data attribution (TDA) is a set of principled methods that aims to quantify the influence of individual training samples on machine learning model predictions (Deng et al., 2025). Most existing TDA methods can be roughly categorized into two types: retraining-based and gradient-based approaches[1] (Hammoudeh & Lowd, 2024; Cheng et al., 2025). *Retraining-based methods*, such as Leave-One-Out (Cook & Weisberg, 1982), Shapley value estimators (Ghorbani & Zou, 2019; Jia et al., 2019; Kwon & Zou, 2021), Data banzhaf (Wang & Jia, 2023), and Datamodels (Ilyas et al., 2022), require training models on multiple data subsets, making them costly and difficult to scale. *Gradient-based influence estimators* typically compute TDA scores by leveraging the gradient signals of training and test instances within the model, estimating how training data affects model predictions, which significantly reduces the computational cost compared to retraining-based approaches. Representative methods include influence function (IF) (Koh & Liang, 2017), representer point selection (RPS) (Yeh et al., 2018), TracInCP (Pruthi et al., 2020), and TRAK (Park et al., 2023). With the development of increasingly efficient and high-performing TDA methods, these techniques have been widely adopted across data-centric AI applications such as data selection (Yu et al., 2024; Engstrom et al., 2024; Coalson et al., 2025), copyright compensation (Deng et al., 2023; Wang et al., 2024a), and fact tracing (Chang et al., 2024; Choenni et al., 2023; Akyürek et al., 2022).

Despite their broad applications, TDA methods face several key challenges in real-world scenarios, where model information and computational resources are often limited. Most TDA approaches assume full visi-

---

[1]Some representation-based approaches like LAVA (Just et al., 2023) are also proposed.

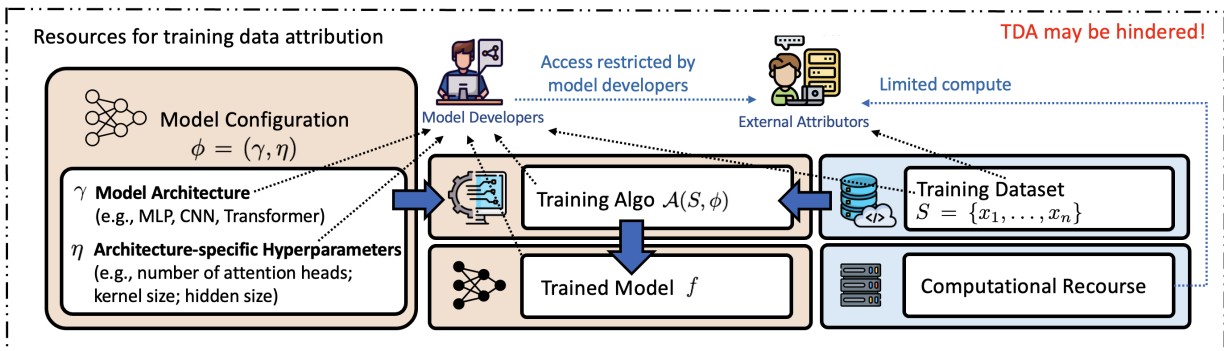

Figure 1: **Illustration of our motivation.** This figure highlights the key resources required for training data attribution, including the model configuration $\phi = (\gamma, \eta)$, where $\gamma$ denotes the high-level model architecture and $\eta$ represents the architecture-specific hyperparameters; the training dataset $S$; the training algorithm $\mathcal{A}$; the trained model $f$; and the computational resources. In practice, data attribution may be hindered by limited access to key resources controlled by model developers, or by insufficient computational resources on the external attributor's side.

bility into critical model components, including not only the exact architecture and trained parameters, but sometimes training configurations such as optimization algorithms and learning rate schedules. In addition, computational resources for retraining a full model are sometimes required. These assumptions could be feasible for small or self-developed models, but are rarely practical for modern generative AI systems, particularly commercial services powered by large models. For example, in applications involving large language models (LLMs) such as GPT-4 (Achiam et al., 2023) and Claude-3.5-Sonnet (Anthropic, 2024), model developers typically do not disclose internal details like model parameters or architectures, and instead only provide access through APIs. Moreover, some external attributors may lack the computational resources required to (repeatedly) retrain models with billions of parameters as required by some TDA methods. These limitations raise a key research question:

> *Can we perform effective training data attribution when model details are inaccessible and computational resources are constrained?*

There have been some existing studies related to the challenges of performing TDA under the aforementioned limited-access constraints. However, these approaches fall short in different ways. LAVA (Just et al., 2023) is a model-agnostic method for data attribution; this very feature limits its ability to attribute data importance to a specific model, as it is more focused on general data valuation. A recent work proposed by Khaddaj et al. (2025) explores the use of small proxy models to substitute the original models. Still, it is primarily motivated by improving attribution efficiency and lacks comprehensive experiments demonstrating how to effectively select or design proxy models. In addition, there have been many efficient TDA methods—such as different variants of influence functions, including CG (Martens et al., 2010), LiSSA (Agarwal et al., 2017), DataInf (Kwon et al., 2023), EK-FAC (Grosse et al., 2023), and LoGra (Choe et al., 2024). These efficient TDA methods still rely on the assumption of having access to at least one full model training.

To fill this gap, we first consider the practical constraints that TDA methods may encounter in real-world scenarios (see Figure 1 for an overview). We formalize several common restricted-access scenarios encountered in practical applications. Specifically, we represent the learning configuration of a model as $\phi = (\gamma, \eta)$, where $\gamma$ denotes the architecture family (e.g., Transformer, ResNet), and $\eta$ represents architecture-specific implementation details or hyperparameter settings (e.g., number of layers, hidden dimensions). Based on the trained target model being not accessible, we consider the following five restricted-access settings: (1) the model architecture $\gamma$ is known and query access is available, but the exact parameter configuration $\eta$ is unknown; (2) the model architecture $\gamma$ is known, but neither query access nor parameter details $\eta$ are available; (3) only query access is granted without knowledge of the architecture or parameters; (4) no information about the target model is accessible; and (5) computational resources are limited. For each of these scenarios, we propose corresponding strategies to mitigate the lack of access or resources, attempting to

enable the implementation of existing TDA methods under these constrained conditions. Note that, in this work, we assume the external attributors have full access to the training dataset, with detailed justification provided in Section 2.3. In brief, attribution requires knowledge of candidate data, training sets are often disclosed for transparency or compliance, and user-provided finetuning is supported by many modern APIs. Building on this framework, we conduct extensive experiments across diverse modalities and data settings. These experiments, as detailed in Section 4, enable a systematic investigation of attribution effectiveness under different access constraints, with a particular focus on understanding the role and impact of proxy model selection in constrained scenarios.

Our results show that access to the target model's architecture is critical for effective proxy-based attribution. When the proxy shares the same architecture family, it can approximate attribution scores well, even with different capacities. In contrast, proxies with mismatched architectures—selected randomly—fail to align, even with query access and distillation signals. Furthermore, we find that even completely untrained models can yield informative attribution results, highlighting the potential of low-cost and efficient alternatives for practical TDA in constrained scenarios. Finally, we note a limitation of the LDS metric: it fails to evaluate attribution performance for a specific model checkpoint, suggesting the need for more targeted evaluation methods in future research.

We summarize our contributions as follows.

- We identify and formalize several practical constraints for applying data attribution, including varying degrees of access limitations and computational resource constraints.
- We design alternative methods to perform data attribution under various restricted-access settings. While our experiments confirm their feasibility in scenarios where partial model information is available, we also identify limitations—particularly, attribution becomes infeasible when no knowledge about the target model architecture is accessible.
- We demonstrate that calculating TDA scores on untrained models can still yield non-trivial performance, offering an efficient alternative in resource-constrained settings.

## 2 Problem Definition

In this section, we formally define our problem setting and describe the motivation and constraints associated with each scenario.

### 2.1 Preliminary

In real-world training data attribution settings, we consider two types of roles: model developers and external attributors. The model developers is the direct owner of the model and thus has full access to the model configuration and training details, which is typically necessary for supporting general data attribution tasks. The external attributors aim to understand how individual training examples contribute to the model's predictions on one or more test inputs but typically do not have direct access to model-related information. For example, in *data compensation* applications, data providers often act as external attributors, seeking to evaluate the relative importance or ranking of their contributed data within the full training set. In such cases, the target commercial model is owned by the corresponding model developers. Therefore, the feasibility of performing data attribution largely depends on the extent to which the model developers discloses information about the model.

**The AI Model.** To better characterize the information accessible to the external attributors, we formalize the AI model training process as follows. Let $S = \{x_1, \ldots, x_n\}$ denote the training dataset, where $S \in \mathcal{S}$ is the space of all possible training sets and each $x_i$ is an input-label pair $(x_i^{\text{feat}}, x_i^{\text{label}})$. Let $\Phi$ denote the space of model configurations, where each configuration $\phi = (\gamma, \eta)$ includes an architecture family $\gamma$ (e.g., Transformer, ResNet) and architecture-specific hyperparameters $\eta$ (e.g., number of layers, hidden size). Let $\mathcal{F}$ denote the set of trained models. A training algorithm $\mathcal{A} : \mathcal{S} \times \Phi \to \mathcal{F}$ maps a training dataset $S$ and a configuration $\phi$ to a trained model $f \in \mathcal{F}$.

In particular, the model developers has access to the trained model $f$. They also possess detailed knowledge of the training algorithm $\mathcal{A}$, including the specific model configuration $\phi$ (i.e., the model architecture and associated hyperparameter settings) used during training. We also define a query function $Q(\cdot)$ as the external interface that returns the model's prediction for any input $x^{\text{feat}}$, that is, $Q(x^{\text{feat}})$ returns the same output as the trained model $f(x^{\text{feat}})$, although its internal implementation may not be accessible. This interface is typically exposed to users by the model developers to support limited forms of model access.

**The TDA Problem.** Let $T = \{z_1, \ldots, z_m\} \in \mathcal{T}$ denote the test dataset, where $\mathcal{T}$ is the space of all possible test datasets. A data attribution method $\tau : T \times \mathcal{S} \times \mathcal{F} \to \mathbb{R}^n$ takes as input a test point $z \in T$, a training dataset $S \in \mathcal{S}$, and a trained model $f \in \mathcal{F}$, and returns a score vector $\tau(z, S; f) \in \mathbb{R}^n$ quantifying the contribution of each $x_i \in S$ to the model's prediction on $z$. Since the trained model $f$ is obtained by applying the training algorithm $\mathcal{A}$ to the dataset $S$ and configuration $\phi$, we may equivalently write $\tau(z, S; \mathcal{A}(S, \phi))$. We provide a table of notation in Appendix E.

## 2.2 TDA Constraint Scenarios

As previously discussed, the extent to which the model developers discloses information about their model can vary, especially in real-world scenarios involving commercial models where stricter limitations are often imposed. Beyond the standard data attribution setting, we begin by assuming that the specific trained target model $f$ is not open-source and thus cannot be directly used or inspected. Given this constraint, we categorize the commonly TDA constraint scenarios into the following types based on what is typically disclosed in practice:

- **Standard TDA Setting: Full Access.** This represents the ideal scenario for implementing TDA methods. The model is open-source, meaning the trained model $f$ can be directly accessed, including its learned internal parameters. In addition, users have full access to the training configuration $\phi = (\gamma, \eta)$, which specifies both the model architecture $\gamma$ and its corresponding hyperparameters $\eta$. With such information, the training process can be reproduced, and internal signals such as gradients are fully available. This enables external attributors to directly implement attribution methods such as TRAK, influence functions, or TracIn. Many well-known open-source models fall into this category. For example, Stable Audio Open (Evans et al., 2025) provide detailed architecture specifications, pre-trained weights, and open-source training code, often accompanied by technical reports or blog posts describing their design and training procedures.

- **Scenario 1: Known Architecture, Query Access.** In this setting, the model developers discloses the high-level model architecture $\gamma$ while keeping the internal hyperparameters $\eta$ undisclosed. Users are allowed to query the model to obtain outputs such as logits or probabilities through the query interface $Q(\cdot)$ defined earlier. A representative example is GPT-3 (Brown et al., 2020), which reveals the overall architecture family, but the specific hyperparameter settings and internal configurations are not publicly available through any official sources.

- **Scenario 2: Known Architecture Only.** In this case, users have access only to the high-level model architecture $\gamma$, but lack knowledge of the specific internal hyperparameter configuration $\eta$. They also cannot query the model or obtain its outputs. As a result, direct access to the model's gradient signals is not possible, and external attributors are unable to verify or analyze the model's behavior through sampling. This scenario typically arises when models are introduced in blog posts or technical reports, where the design philosophy, component structure, or training objectives may be described, but no APIs, pretrained weights, or implementation details are released.

- **Scenario 3: Query Access Only.** Under this condition, the model developers only allow users black-box access to the model outputs, that is, they can only obtain a data response through the query function $Q(\cdot)$, but have no information about the model architecture $\gamma$ or hyperparameters $\eta$, making it impossible to reproduce the training algorithm $\mathcal{A}(S, \phi)$. This situation is typical for commercial models such as Suno AI (Suno AI).

- **Scenario 4: No Access to the Target Model.** In the most restrictive setting, users cannot access the target model's configuration $\phi$, the trained model $f$, nor can they issue any queries via $Q(\cdot)$. External users may only have coarse-grained knowledge of the model's functionality or high-level objective. This scenario typically arises when the model is deployed within proprietary systems, such as financial risk engines or internal monitoring platforms. For commercial or security reasons, such systems often do not expose any form of interface or model information to external users.

- **Scenario 5: Attribution Without Training.** In some cases, users have full knowledge of the target model's configuration $\phi = (\gamma, \eta)$ but have not yet trained or obtained a trained model $f$. Traditional TDA methods often require one or multiple trained model to compute attribution scores such as TDA ensemble. The model training could be computationally heavy and hinder external attributors to perform TDA. If TDA scores could be estimated without requiring model training, it would significantly improve the practicality of the technique and facilitate applications such as early-stage data curation by reducing overhead before a model is trained.

| Information Category | Notation | Description |
|---|---|---|
| $I_1$: Target Model Full Configuration | $\phi = (\gamma, \eta)$ | Full training configuration, including model architecture and associated hyperparameters. |
| $I_2$: Model Architecture Family | $\gamma$ | Main Architecture (e.g., ResNet, Transformer). |
| $I_3$: Architecture-specific Hyperparameters | $\eta$ | Architecture-dependent parameters (e.g., number of layers, hidden size). |
| $I_4$: Trained Model | $f$ | Attribution target model after training. |
| $I_5$: Query Function | $Q(\cdot)$ | Model's black-box interface that maps input queries to output responses. |

| Constraint Scenario | Accessible Info | Representative Examples |
|---|---|---|
| $S_0$: Full Access (Standard TDA) | $I_1 = (I_2 + I_3), I_4, I_5$ | BERT, ResNet, Stable Audio Open. |
| $S_1$: Known Architecture + Query Access | $I_2, I_5$ | GPT-3. |
| $S_2$: Known Architecture Only | $I_2$ | Models described only via blogs or technical reports. |
| $S_3$: Query Access Only | $I_5$ | Suno AI (API-only access). |
| $S_4$: No Access | $-$ | Internal enterprise models (e.g., finance, surveillance systems). |
| $S_5$: Limited Compute | $I_1 = (I_2 + I_3)$ | Large-scale models before training, or TDA for pre-training data selection. |

Table 1: **Overview of TDA resource access and constraint scenarios.** The upper portion enumerates types of information relevant to TDA. The lower portion categorizes practical constraint settings based on accessible information combinations and list their representative examples.

### 2.3 Training Data Transparency.

Apart from model information access, data is another critical aspect for data attribution. However, missing data access will largely (and more severely) break the data attribution task definition we defined in Section 2.1 that TDA quantifies the contribution of individual training examples. *Membership inference attack* (Hu et al., 2022) is a task that is more commonly referred to when data access is limited, which is beyond the scope of this paper.

Besides, ongoing legislative efforts to enhance the transparency of AI training data are steadily advancing. For example, the EU's AI Act of 2024[2] in the European Union asks providers of general-purpose AI models to publish detailed summaries of their training data. Furthermore, several *proposals*, like the AI Foundation Model Transparency Act of 2023[3] and the Generative AI Copyright Disclosure Act of 2024[4] require companies to notify related government organizations about their data sources. Based on this evolving regulatory landscape, we do not discuss the cases where data access is limited in this work. Still, we highlight here that it could be another interesting topic to discuss.

## 3  Method

This section explores attribution strategies aimed at mitigating the challenges posed by limited access to model information. We propose corresponding strategies to the resource-restricted scenarios discussed in Section 2.2. Figure 2 provides an overview of these strategies and their applicable conditions.

---

[2] https://eur-lex.europa.eu/eli/reg/2024/1689/oj/eng
[3] H.R.6881
[4] H.R.7913

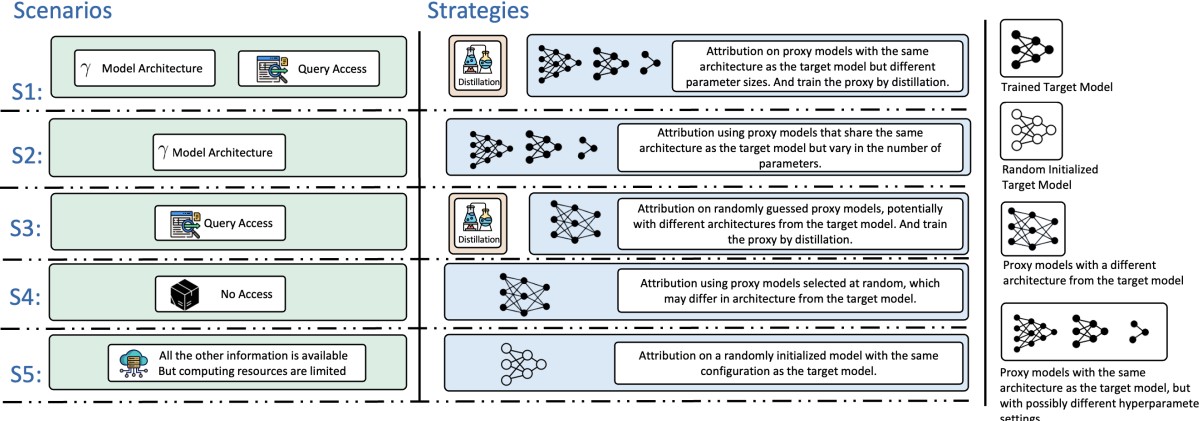

Figure 2: **Attribution strategies under different resource-constrained scenarios.** The right side of the figure provides a detailed explanation of the models mentioned in the strategies. The distillation part in the method refers to distillation using a trained target model.

## 3.1 Proxy Modeling with Known Architecture

This strategy corresponds to Scenario 1 and Scenario 2 introduced in Section 2.2. In these scenarios, users have access to the high-level architecture of the target model but may not know the corresponding hyperparameters. To enable gradient-based data attribution under such constraints, a natural approach is to train a proxy model on the same dataset as the target model. Gradient-based TDA methods can then be applied on the proxy model to approximate the gradient signals that would be obtained from the original model. Intuitively, the closer the proxy model is to the target model, the more reliable the resulting attribution estimates are likely to be. Under this partially informed setting, users can construct an estimated model configuration $\phi' = (\gamma, \eta')$ by combining the known architecture $\gamma$ with a plausible guess of the internal hyperparameters $\eta'$. A proxy model $f'$ is then trained using a selected training algorithm $\mathcal{A}'$ on the same dataset $S$, such that $f' = \mathcal{A}'(S; \phi')$. However, a key limitation of this approach is that even when the architectures are similar, the proxy model $f'$ may still deviate significantly from the target model in terms of functional behavior—due to differences in parameter initialization, model capacity, training dynamics, or other unobserved hyperparameters.

**Knowledge distillation.** In Scenario 1, users additionally have access to a query function $Q(\cdot)$ that returns the target model's output given an input. This query function could provide an alignment signal for training the proxy model. Intuitively, we would like the proxy to better approximate the target model's behavior, and to achieve this, we can construct a knowledge distillation framework (Hinton et al., 2015), where the target model serves as the teacher and the proxy model as the student. The training objective for the proxy combines the standard supervised loss with a distillation loss that encourages it to match the output (e.g., the output logits) of the teacher model:

$$\mathcal{L}_{\mathrm{KD}} = \alpha T^2 \cdot \mathrm{KL}\left(\mathrm{softmax}\left(\frac{Q_{\mathrm{stu}}(\cdot)}{T}\right) \,\|\, \mathrm{softmax}\left(\frac{Q_{\mathrm{tch}}(\cdot)}{T}\right)\right) + (1 - \alpha) \cdot \mathcal{L}_{\mathrm{sup}},$$

where $Q_{\mathrm{stu}}(\cdot)$ and $Q_{\mathrm{tch}}(\cdot)$ are the logits produced by the student (proxy) and teacher (target) models, respectively. The temperature parameter $T$ controls the smoothness of the probability distributions: a higher $T$ produces softer distributions that expose more fine-grained similarities between classes. The KL divergence measures the difference between the softened output distributions. The $\alpha \in [0, 1]$ balance the distillation loss with the standard supervised loss $\mathcal{L}_{\mathrm{sup}}$ (e.g., cross-entropy or language modeling loss). This setup allows the proxy model to better approximate the decision boundaries of the target model, thereby improving the reliability of attribution results computed on the proxy. However, it is important to note that even if two models share very similar decision boundaries, their gradient vectors on the same data points

may still differ significantly, which means that the resulting TDA outcomes may not necessarily become more aligned.

## 3.2 Proxy Modeling with Unknown Architecture

This strategy corresponds to Scenario 3 and Scenario 4, where users have no access to the model's architecture $\gamma$ or its hyperparameters $\eta$. In this case, users must rely on heuristics or information related to the target task to hypothesize a plausible model configuration $\phi' = (\gamma', \eta')$, and train a proxy model $f' = \mathcal{A}'(S; \phi')$ on the same dataset $S$. When the guessed proxy architecture $\gamma'$ differs significantly from the target model architecture $\gamma$, the discrepancy between the proxy and target models becomes more pronounced.

In Scenario 3, users retain access to the target model's query function and can therefore apply knowledge distillation to align the proxy model's predictions with those of the target. In Scenario 4, however, no query access is available. As a result, the proxy model must be trained solely on the training dataset, without any auxiliary supervision from the target model's outputs.

## 3.3 Attribution without Model Training

This final strategy corresponds to Scenario 5, where the training procedure and model configuration $\phi = (\gamma, \eta)$ are assumed to be fully known, but the user wishes to completely avoid the computational overhead of training. This setting typically arises in large-scale model applications. Given that the full configuration of the target model is available, one possible strategy is to use a randomly initialized model with the same configuration. Additionally, if the target model is fine-tuned from a widely used pre-trained backbone (e.g., BERT or GPT-2), users may leverage the vanilla checkpoint prior to fine-tuning, which requires no additional computational cost. We admit this strategy departs from the theoretical assumptions underlying many data attribution methods, such as the influence function, which assumes that the model is trained to convergence. However, one may still expect this strategy to be effective in practice, as the gradients computed from a randomly initialized or pre-trained model can reflect directions for reducing the loss, thereby providing meaningful signals for data attribution.

# 4 Experiments and Discussions

In this section, we conduct comprehensive experiments to evaluate the performance and behavior of TDA under various resource-constrained scenarios, with strategies outlined in Section 3. We explore their effectiveness across different tasks and settings and present the quantitative attribution results and corresponding analysis under different constraints.

## 4.1 Evaluation Metrics

We perform both task-specific and application-agnostic evaluation. Mainstream application-agnostic metrics, such as Leave-One-Out (LOO) correlation and the Linear Datamodeling Score (LDS), are widely accepted counterfactual prediction metrics in the research community (Koh & Liang, 2017; Park et al., 2023; Bae et al., 2024; Choe et al., 2024; Deng et al., 2024). In this work, we primarily focus on LDS, which is a counterfactual evaluation framework introduced by Park et al. (2023). LDS can be formally defined as the equation below.

**Definition 4.1 (Linear Datamodeling Score)** *Let $S$ denote the full training dataset, and $\tau$ denote a data attribution method. Consider $m$ randomly sampled subsets $\{S_1, \ldots, S_m\}$, each of size $\alpha \times n$ for some fixed $\alpha \in (0, 1)$, where $n$ is the size of the full train set $S$. The linear datamodeling score (LDS) of $\tau$ for a given test example $z$ is defined as the Spearman rank correlation (Spearman, 1961) between model predictions and attribution-based predictions:*

$$\mathrm{LDS}(\tau, z) \triangleq SpearmanCorr\left(\{f_{S_j}(z)\}_{j=1}^m, \ \{\sum_{i:x_i \in S_j} \tau(z, S_j; f)_i\}_{j=1}^m\right),$$

where $f_{S_j}$ is the model output function trained on subset $S_j$.

Though such application-agnostic metrics are widely used in TDA studies, we also perform the evaluation using several application-driven metrics, such as noisy label detection and data selection to provide a more comprehensive evaluation. For noisy label detection, we evaluate performance using the Area Under the ROC Curve (AUC). Besides, we also include the brittleness test (Ilyas et al., 2022), another counterfactual prediction-based evaluation method that measures the number of most influential training examples that need to be removed to flip the prediction of a given test input. A more detailed explanation is provided in Appendix F.

### 4.2 Experiment 1: Attribution via Proxy Models under Information Constraints

**Experiment Design.** We select three different experimental settings to represent the limited-access settings provided by model developers. For each target model, we construct a corresponding group of proxy models, which are trained on the same dataset as the target model and used to compute TDA scores. The selection and training of proxy models follow the strategies described in Sections 3.1 and 3.2. In particular, we include a proxy model that is architecturally identical to the target model, serving as a baseline for the "full-access" or "perfect-guess" setting. To evaluate the effectiveness of a proxy model $f'$ in performing data attribution under the operational setting of $f$, we adopt LDS with ground truth derived from the target model. Specifically, we sample $m = 50$ random subsets $\{S_j\}_{j=1}^m$ from the training set and retrain the target model $f_{S_j}$ on each subset [5]. This LDS with model f ground truth is defined as $\text{LDS}_f(\tau, z) = \text{SpearmanCorr}\big(\{f_{S_j}(z)\}_{j=1}^m, \{\sum_{x_i \in S_j} \tau(z, S_j; f')_i\}_{j=1}^m\big)$, where $f_{S_j}$ denotes the target model trained on subset $S_j$.

| Setting | Modality | Task | Dataset | Target Model | Sample Size (Train, Test) |
|---|---|---|---|---|---|
| (1) | Image | Classification | CIFAR-2 (Krizhevsky et al., 2009)[6] | ResNet-9 (He et al., 2016) | (5000, 500) |
| (2) | Music | Generation | MAESTRO (Hawthorne et al., 2018) | MusicTransformer (Huang et al., 2018) | (5000, 178) |
| (3) | Text | Generation | WikiText-2 (Merity et al., 2016) | GPT-2 (Radford et al., 2019) | (4656, 481) |

Table 2: **Experimental setups including modalities, tasks, datasets, and target models.**

**Models, Datasets, and Attribution Methods.** We evaluate the attribution performance of proxy models under various settings, as summarized in Table 2, covering three modalities: image, music, and text. The choice of proxy models is guided by the task of the corresponding target model to better reflect realistic scenarios. Detailed configurations of the proxy models and training process are provided in Appendix A.2. In this experiment, we primarily evaluate the attribution performance of TRAK (Park et al., 2023), one of the state-of-the-art data attribution methods.

**Experiment Results.** From Table 3, we draw two key conclusions. **(1) Data attribution is feasible when the proxy model shares a similar architecture with the target model, but mismatches in model structure, despite comparable parameter sizes, can still result in substantial degradation in attribution quality.** Across all three experimental settings, we observe that proxy models with architectures similar to the target yield attribution scores similar performance in terms of LDS. In the ResNet-9 setting, variants such as ResNet-9-Deep, ResNet-9-Wide, and ResNet-9-Small achieve comparable attribution performance despite differences in parameter count, due to their architectural similarity. Even TinyResNet, which has only 0.015× the parameters of the target model, produces meaningful attribution scores. This effectively simulates Scenario 1 and Scenario 2, where the high-level architecture of the target model is known; in such cases, random guessing of hyperparameters does not significantly affect attribution quality, indicating that both scenarios are feasible. Notably, although ResNet-9-Deep and ResNet-18 have similar parameter sizes, ResNet-9-Deep achieves significantly better attribution performance due to closer architectural alignment [7] with the target ResNet-9. The corresponding LDS scores (from models trained from

---

[5]All 50 trained instances are obtained from the *dattri* library (Deng et al., 2024) to ensure reproducibility.

[6]CIFAR-2 is a binary classification subset of CIFAR-10 (cats vs. dogs).

[7]ResNet-9-Deep doubles the number of BasicBlocks in each residual layer, increasing the network depth while preserving the overall architectural pattern of ResNet-9. Additional details on proxy model architectures can be found in Appendix A.2.

| Proxy Model | #Params | Params Ratio | LDS | |
|---|---|---|---|---|
| | | | TRAK-Train from Scratch | TRAK-KD |
| *ResNet-9 as Target Model* | | | | |
| ResNet-18 | 11.17M | 2.31× | 0.0994 | 0.1050 |
| ResNet-9-Deep | 11.02M | 2.28× | 0.2773 | 0.2695 |
| ResNet-9-Wide | 7.56M | 1.57× | 0.3242 | 0.3338 |
| **ResNet-9-Base** | 4.83M | 1.00× | 0.3235 | 0.3316 |
| ResNet-9-Small | 1.21M | 0.25× | 0.2835 | 0.3124 |
| TinyResNet | 72.85K | 0.015× | 0.2118 | 0.2077 |
| CNN | 267.62K | 0.055× | 0.1367 | 0.1181 |
| MLP | 196.80K | 0.041× | 0.0515 | 0.0661 |
| Logistic Regression | 6.15K | 0.0013× | 0.0541 | 0.0541 |
| *Music Transformer as Target Model* | | | | |
| Music Transformer - 9 Layer | 19.47M | 1.49× | 0.2872 | 0.2996 |
| **Music Transformer - 6 Layer** | 13.12M | 1.00× | 0.4578 | 0.4557 |
| Music Transformer - 3 Layer | 6.72M | 0.52× | 0.4480 | 0.4540 |
| *GPT-2 as Target Model* | | | | |
| GPT-2 Medium | 345.52M | 2.78× | OOM | OOM |
| **GPT-2** | 124.44M | 1.00× | 0.1689 | 0.1668 |
| DistilGPT2 | 81.91M | 0.66× | 0.1645 | 0.1676 |
| TinyStories-GPT2-3M | 3.68M | 0.03× | 0.1039 | 0.1003 |

Table 3: **Comparison of LDS results using different proxy models and training strategies.** *#Params* refers to the total number of trainable parameters in the corresponding model, and *Params Ratio* indicates the ratio of this model's parameter number to that of the target model. The columns *Train from Scratch* and *KD* represent two strategies for training the proxy models. *Train from Scratch* refers to training the proxy model directly on the target dataset from random initialization, while *KD* denotes knowledge distillation from the target model. Distillation configurations are detailed in Appendix A.2. The LDS scores are obtained by using the target model as ground truth to evaluate the attribution quality of TRAK scores computed on the proxy models.

scratch) are 0.2773 and 0.0994, respectively. As the architectural differences become more pronounced—such as when using LR or CNN as proxies—the LDS scores deteriorate accordingly. This corresponds to Scenario 3 and Scenario 4, where the choice of proxy architecture must rely solely on the target task; when the chosen architecture differs substantially from the true one, attribution performance drops significantly. **(2) Knowledge distillation does not significantly enhance attribution performance.** In our experiments, we only performed experiments on distilled proxy models that produced output distributions on the test set that were closer to the target model, with lower KL divergence compared to their non-distilled counterparts. This indicates that the distillation process was successful. However, the attribution performance remained largely unchanged. This suggests that for structurally different models, improved output similarity does not necessarily lead to better gradient alignment or attribution performance.

We further identify a limitation of the LDS evaluation framework. In our study, LDS effectively measures whether attribution results from proxy models approximate those under the target model configuration, but this differs from evaluating whether attribution computed on the proxy model can replicate the attribution results from a specific trained model. LDS is designed to assess attribution methods under a class of model configurations and training algorithm $\mathcal{A}$, which could involve randomness. However, many attribution methods—such as TracIn (Pruthi et al., 2020), SGD-Influence (Hara et al., 2019), and Data Value Embedding (Wang et al., 2024b)—are inherently designed to explain model behavior from a single training trajectory. Applying a general-purpose evaluation like LDS to such instance-specific methods may deviate from their original purpose. This highlights the potential value of **developing attribution evaluation metrics that are application-agnostic yet tailored to individual trained models**.

### 4.3 Experiment 2: Attribution without Model Training under Computational Constraints

For Scenario 5 in Section 2.2, we explore an extreme case of performing TDA under limited computational resources. As discussed in Section 3.3, we attempt to conduct TDA using either a randomly initialized model or a pretrained model without any task-specific fine-tuning.

### 4.3.1 Attribution on Random Initialized Models

To investigate the effectiveness of data attribution on untrained models, we first conduct a preliminary study using fully randomly initialized models to compute attribution scores and assess their quality. We evaluate four representative data attribution methods: influence function (IF) (Koh & Liang, 2017), TracIn (Pruthi et al., 2020), representer point selection (RPS) (Yeh et al., 2018), and TRAK (Park et al., 2023). The experiments are conducted under three settings: a logistic regression model trained on MNIST, a three-layer MLP trained on MNIST, and a ResNet-9 model trained on CIFAR-2. Evaluation metrics include the LDS and the AUC metric described above. For LDS evaluation, each experiment uses a dataset comprising 5,000 training examples and 500 test examples. For AUC evaluation, 10% of the labels are randomly selected and flipped in a 1,000 sample dataset.

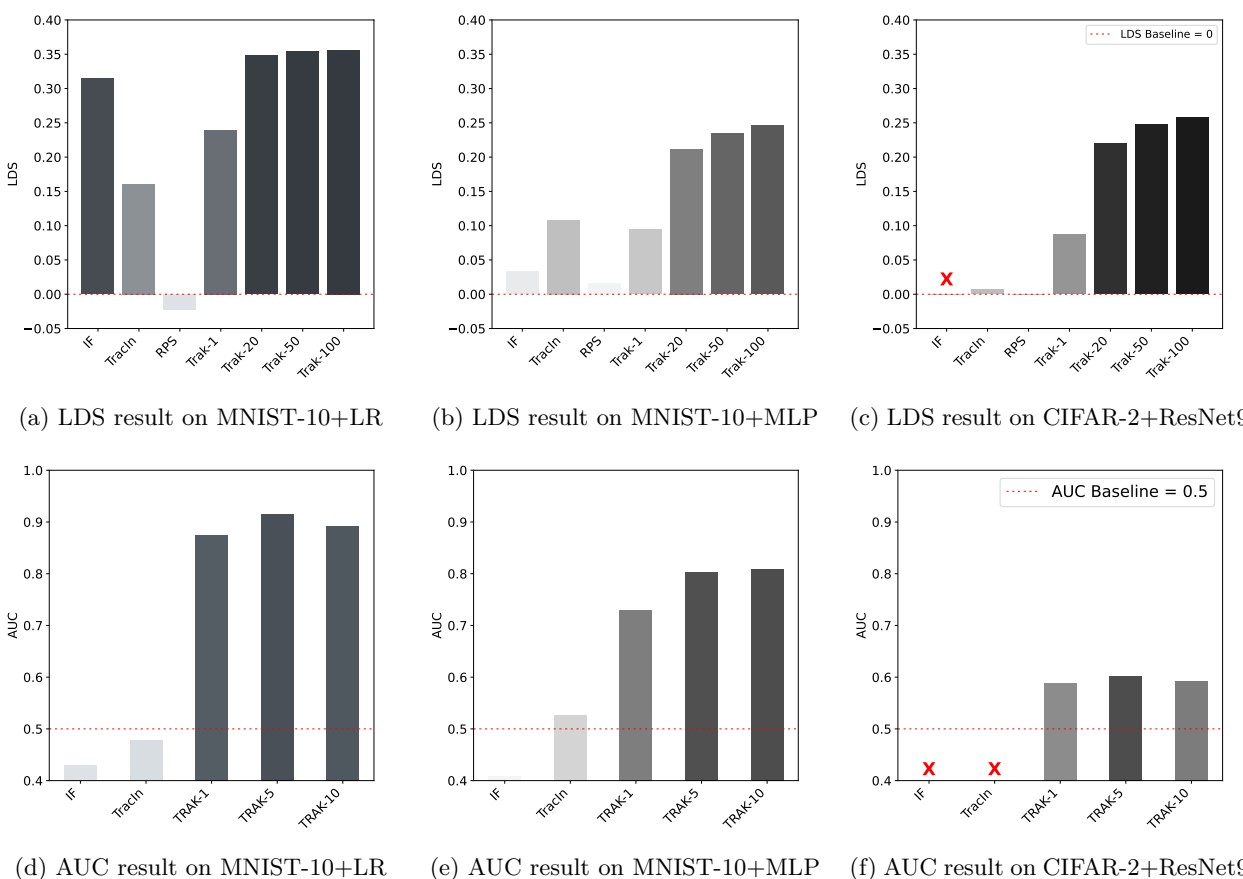

(a) LDS result on MNIST-10+LR    (b) LDS result on MNIST-10+MLP    (c) LDS result on CIFAR-2+ResNet9

(d) AUC result on MNIST-10+LR    (e) AUC result on MNIST-10+MLP    (f) AUC result on CIFAR-2+ResNet9

Figure 3: **LDS and AUC evaluation results on randomly initialized models.** X mark means the out-of-memory issue. The numeric suffix of each TRAK variant indicates the number of ensembles used. The dotted line indicates the baseline where LDS equals 0 and AUC equals 0.5. For each bar, darker colors represent better performance.

**Results.** As shown in Figure 3, TRAK consistently demonstrates non-trivial results in both the LDS and AUC evaluation experiments. Specifically, the LDS values are significantly greater than zero, and the AUC scores exceed 0.5. Moreover, the attribution scores produced by TRAK become better as the ensemble size increases, similar to the behavior of TRAK performance on trained models. For IF and TracIn, both methods also exhibit non-trivial results, with LDS values above zero and AUC scores exceeding 0.5. However, their performance remains weaker than TRAK, possibly due to their inherently lower attribution quality even on trained models. In contrast, RPS performs close to random. **These two sets of experimental results are sufficient to demonstrate that, particularly for TRAK, informative data attribution scores can still be obtained even when the model is randomly initialized.**

### 4.3.2 Scaling up to Foundation Models with Fine-tuning Setting

Language models are often fine-tuned on specific datasets starting from a pretrained model. We aim to investigate whether data attribution scores computed solely on the pretrained model already carry meaningful information. We adopt the experimental setting of training GPT-2 on the WikiText-2 dataset, and evaluate the informativeness of attribution scores via a data selection task. We compute training data attribution scores with respect to the evaluation set using both the pre-trained (unfine-tuned) GPT-2 model and the fine-tuned GPT-2 model. Based on these scores, we filter the training dataset by retaining only the samples with top-60% attribution scores. We then fine-tune the vanilla GPT-2 model on this filtered subset and evaluate the quality of the selected data by tracking the loss on the evaluation set.

**Results.** As shown in Figure 4, models fine-tuned on subsets selected by TRAK attribution scores outperform the random baseline in terms of evaluation loss. Notably, the subset selected using attribution scores computed from the pre-trained GPT-2 model yields training performance comparable to that of the subset selected using the fine-tuned model. **This indicates that attribution scores computed solely from the pretrained model already carry meaningful information for identifying valuable training examples.** The result highlights the potential of leveraging attribution scores for data selection even before any task-specific fine-tuning, thereby reducing computational overhead. In Table 4, we also present the LDS scores computed using both models. As mentioned before, one possible explanation is that the gradients computed from the untrained model still reflect directions in the parameter space. Although the model has not been adapted to the task, these gradient directions can still identify influential training points.

### 4.4 Discussions

Our experiments demonstrate that training data attribution remains feasible under limited-information settings when the high-level architecture of the target model is known. Even without access to detailed architectural hyperparameters, attribution results via proxy models with similar architectures remain effective even when the proxy and target models differ in parameter scale. In contrast, when the architecture is completely unknown, using heterogeneous proxy models could lead to substantial degradation in attribution performance despite access to model outputs, highlighting the critical role of architectural priors in this task. Another key insight is that attribution can be informative even before training begins. Attribution scores derived from randomly initialized or only pretrained language models still exhibit meaningful patterns.

| Model | LDS Score |
|---|---|
| Pre-trained GPT-2 | 0.1821 |
| Fine-tuned GPT-2 | 0.1613 |

Table 4: **Comparison of LDS with target model ground truth (fine-tuned GPT-2) scores for TRAK attribution under different model states.** Notably, the attribution scores obtained from the unfine-tuned GPT-2 model slightly surpass those of the fine-tuned model, despite having no exposure to the target dataset during training.

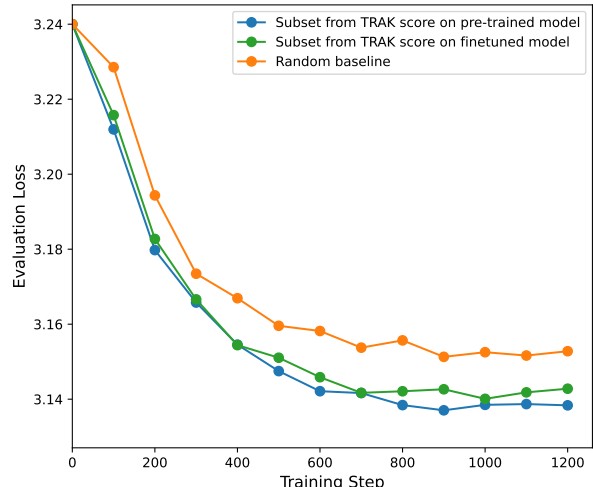

Figure 4: **Evaluation loss curves during fine-tuning on subsets selected by different attribution scores.** The subsets are selected based on TRAK scores computed using (1) the pre-trained GPT-2 model, (2) the fine-tuned GPT-2 model, and (3) a random score.

## 5 Related Work

**Proxy Model for Data Selection and Data Attribution.** Some existing studies have shown that small models can exhibit behavior consistent with large models under certain specific conditions (Yang et al., 2023;

Vyas et al., 2023). This trait makes small-scale proxy models a useful tool for data pre-filtering, data cleaning, and other tasks in the current data-centric paradigm (Xie et al., 2023; Yu et al., 2024), due to their lower parameter count and higher computational efficiency. Recent studies have also explored the feasibility of using proxy models for TDA. For example, Engstrom et al. (2024) shows that even Datamodels trained with small models (e.g., 125M) can provide effective data selection for training much larger models (e.g., 1.3B). Khaddaj et al. (2025) takes this further by showing that even proxy models trained with up to $175\times$ or $370\times$ less compute than the target large-scale models can effectively approximate the data attribution results produced by the large models using TRAK. While prior work has largely emphasized computational efficiency through proxy models, how to select suitable proxies for TDA in settings where the target model is inaccessible or not well specified has not been well explored before this work.

**Alternative Attribution Methods under Limited Access Constraints.** Previously, some studies have explored TDA without involving specific learning algorithms, such as the LAVA framework proposed by Just et al. (2023), which estimates data value using a class-wise Wasserstein distance in a learning-agnostic manner. These approaches aim to assess the intrinsic quality of data rather than its impact on specific model decisions. Besides, measuring the similarity between data points also does not rely on model-specific information and huge computational costs. Some prior TDA works have leveraged similarity-based techniques, such as TF-IDF matching, as studied in Grosse et al. (2023), or k-Nearest Neighbors (kNN) (Guo et al., 2020)—to perform pre-selection of candidate data points before applying full attribution methods.

However, when the goal is to implement data compensation or data selection using TDA methods, the external attributors typically requires a fair assessment of how each data point influences a specific model's behavior. In contrast, the alternative methods discussed above primarily focus on evaluating the intrinsic quality or pairwise similarity of data points, rather than their actual impact on model decisions. While these learning algorithm-agnostic approaches are related in spirit and offer certain advantages, their utility remains limited when precise, model-specific attribution is required. Therefore, they do not align with the core objective of our research.

## 6 Conclusion and Future Work

In this work, we conduct a systematic investigation of training data attribution (TDA) under varying information and resource constraints, with a focus on scenarios where model access is restricted and training resources are limited. Through experiments across multiple modalities and model architectures, we demonstrate that proxy-based TDA can serve as an effective alternative when direct access to the target model is unavailable. We find that architectural alignment between the proxy and target model is more important than model size, and attribution results are significantly improved when the two models share the same architecture family. Therefore, the selection strategy of the proxy model should aim to match the target model's architecture family as closely as possible. Additionally, we show that TDA methods can produce informative results even on untrained models, which could be helpful for some downstream tasks such as performing preliminary data filtering with minimal training cost.

**Limitations and future work.** Our findings suggest several directions for future research. First, there is a lack of effective evaluation metrics for assessing attribution quality on specific trained models. Existing metrics such as LDS focus on evaluating datasets across model families, rather than attribution for a fixed model. While some studies attempt attribution during training, their evaluations often rely on downstream task performance instead of direct, quantifiable measures. Second, we observe that current methods still fail to support data attribution for fully black-box models—those where only output predictions are accessible and no internal details are available. Future work should explore the development of attribution techniques that can operate under such strict access constraints, enabling fair and transparent data attribution even in closed-source, commercial AI systems. Third, we assume that the training data can be fully accessed by the external attributors, which is still not the case for many generative AI models.

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

# A Experiments Details.

## A.1 Dataset Licenses and Computational Resources

We use the following publicly available datasets in our experiments: The MNIST-10 dataset is licensed under the CC BY-SA 3.0 license. The CIFAR-2 dataset is released under the MIT license. The WikiText-2 dataset is also licensed under the CC BY-SA 3.0 license. The MAESTRO dataset is licensed under the CC BY-NC-SA 4.0 license. All the experiments are conducted on `NVIDIA A100` GPUs with `40 GB memory` and `NVIDIA A40` GPUs with `48 GB` memory.

## A.2 Target Model and Proxies Details

**ResNet-9 and corresponding proxy models**  For the setting where ResNet-9 is used as the target model, we construct a series of proxy models derived from it. These include: `ResNet-9-Wide`, which increases the number of output channels in each residual stage while keeping the overall depth unchanged; `ResNet-9-Deep`, we do not change the original number of layers in ResNet-9, but we double the number of BasicBlock units stacked in each layer; `ResNet-9-Small`, which reduces the number of channels in each layer while maintaining the same layer number; and `TinyResNet`, which simplifies the architecture by removing the third layer of ResNet-9 to reduce depth. In addition, we include a deeper variant, `ResNet-18`, which consists of four residual stages, each comprising two BasicBlock modules. We also include a group of non-residual baselines, including a standard convolutional neural network (`CNN`), a three-layer multilayer perceptron (`MLP`), and `logistic regression`, to evaluate the attribution behavior across architectures without residual connections. All models are trained for 20 full epochs. For models utilizing knowledge distillation, we adopt a configuration with an interpolation factor of $\alpha = 0.99$ and a distillation temperature of $T = 2$.

**MusicTransformer and corresponding proxy models.**  We adopt the Music Transformer architecture proposed by Huang et al. (2018) as the target model. The implementation can be found in *dattri* library (Deng et al., 2024). The target model consists of 6 Transformer layers, each with 8 attention heads, a hidden dimensionality of 512, and a feedforward network dimension of 1024. A dropout rate of 0.1 is applied throughout the network, and input sequence length of 2048 tokens. For the proxy models, we retain the overall architecture but modify the number of Transformer layers. Specifically, we construct two variants with 3 and 9 Transformer layers, respectively, while keeping all other hyperparameter identical to those of the target model. All models are trained for three full epochs. For knowledge distillation, we adopt a standard setting with an interpolation coefficient of $\alpha = 0.9$ and a distillation temperature of $T = 2$.

**GPT-2 and corresponding proxy models.**  We set `GPT-2`(Radford et al., 2019) as the target model, which is pretrained and obtained from the Hugging Face model hub, and we follow its original training configurations without modification. For the proxy models, we select models of varying scales that

maintain a similar architecture while differing in model capacity and pretraining data. Specifically, we include `GPT-2 Medium`(Radford et al., 2019), a larger variant of GPT-2 with more Transformer layers and increased hidden dimensionality; `DistilGPT2`(Sanh et al., 2019), a 6-layer distilled version of GPT-2; and `TinyStories-GPT2-3M`(Bird, 2024), a lightweight model with approximately 3 million parameters pretrained on a simplified story corpus. All proxy models are also obtained from Hugging Face. During fine-tuning, each proxy model is trained for 3 full epochs. For models involving knowledge distillation, we follow the standard setting with interpolation weight $\alpha = 0.9$ and temperature $T = 2$.

## B  TDA Method Formulations

**Tracing with Random Projection after Kernel (TRAK).**  TRAK is proposed by Park et al. (2023). It computes attribution scores by averaging the contributions from multiple models $\{f_{S_i}\}_{i=1}^M$, each trained on a randomly sampled subset $S_i \subset S$. For each model $f_{S_i}$, the gradient representations of the training and test samples are linearized and projected for computational efficiency. The final attribution score for a test input $z$ is denoted as:

$$\tau_{\text{TRAK}}(z, S; \{f_{S_i}\}_{i=1}^M) = \frac{1}{M} \sum_{i=1}^M Q_{f_{S_i}} \left( \frac{1}{M} \sum_{i=1}^M \phi_{f_{S_i}}(z)^\top \left( \Phi_{f_{S_i}}^\top \Phi_{f_{S_i}} \right)^{-1} \Phi_{f_{S_i}}^\top \right),$$

where $\phi_{f_{S_i}}(z)$ and $\Phi_{f_{S_i}}$ are the projected gradients of the test and training samples under model $f_{S_i}$, respectively. $Q_{f_{S_i}}$ is a diagonal matrix capturing the model uncertainty for training samples in $S_i$, where each diagonal entry is equal to one minus the correct-class probability. In our experiments, all models are trained on 50% subsamples of the original training set. And $M$ denotes the ensemble size, i.e., the number of such models.

**Influence Function (IF).**  The influence function, introduced by Koh & Liang (2017), could be defined as:

$$\tau_{\text{IF}}(z, S; f) = \left[ \nabla_\Theta \ell(\hat{\Theta}, x_j)^\top H_{\hat{\Theta}}^{-1} \nabla_\Theta \ell(\hat{\Theta}, z) : x_j \in S \right],$$

where $\nabla_\Theta \ell(\hat{\Theta}, z)$ and $\nabla_\Theta \ell(\hat{\Theta}, x_j)$ are the gradients of the loss function with respect to the model parameters $\Theta$, evaluated at the test point $z$ and the training point $x_j$, respectively. $H_{\hat{\Theta}}^{-1}$ denotes the inverse of the Hessian of the empirical training loss, computed at the model parameters $\hat{\Theta}$.

**TracIn with Checkpoints (TracInCP).**  TracInCP, proposed by Pruthi et al. (2020), estimates the influence of a training sample on a test prediction by accumulating the gradient similarity across multiple model checkpoints during training. The influence score for a test input $z$ is defined as:

$$\tau_{\text{TracInCP}}(z, S; \{f^{(i)}\}_{i=1}^I) = \left[ \sum_{i=1}^I \eta_i \cdot \nabla_{\Theta^{(i)}} \ell(\Theta^{(i)}, x_j)^\top \nabla_{\Theta^{(i)}} \ell(\Theta^{(i)}, z) : x_j \in S \right],$$

where $\nabla_{\Theta^{(i)}} \ell(\Theta^{(i)}, z)$ and $\nabla_{\Theta^{(i)}} \ell(\Theta^{(i)}, x_j)$ denote the gradients of the loss function with respect to the model parameters $\Theta^{(i)}$, evaluated at the test point $z$ and the training point $x_j$, respectively, using the model at checkpoint $i$. The scalar $\eta_i$ corresponds to the learning rate used between checkpoints $(i-1)$ and $i$. In this work, we use the final checkpoint of training as a single evaluation point for computing TracIn scores.

**Representer Point Selection (RPS).**  RPS Yeh et al. (2018) explains neural network predictions as a linear combination of training examples via kernel similarity. Assuming the model can be expressed as $f = \Theta_1 h_j$, where $\Theta_1$ is the final linear layer and $h_j$ is the intermediate feature representation, the training objective is:

$$\min_\Theta \left\{ \frac{1}{n} \sum_{j=1}^n \mathcal{L}(x_j, \Theta) + \lambda \|\Theta_1\|_2^2 \right\}.$$

Given a stationary solution $\Theta^*$, the influence of each training point $x_j \in S$ on the test input $z$ is computed as:

$$\tau_{\text{RPS-L2}}(z, S; f) = \left[ \frac{1}{-2\lambda n} \frac{\partial \mathcal{L}(z, \Theta^*)}{\partial \Phi(x_j, \Theta^*)} \cdot h_j^\top h : x_j \in S \right],$$

where $h_j$ and $h$ denote the feature vectors of $x_j$ and $z$, respectively. This approach quantifies influence via inner product similarity in the feature space.

## C   Hyperparameter Selection for TDA Methods

Hyperparameter selection is crucial for the attribution quality of TDA methods (Wang et al., 2025). Table 5 documents the TDA-specific hyperparameter configurations used across our experiments.

| TDA Method | Model | Dataset | Hyperparameters |
|---|---|---|---|
| TRAK | ResNet-9 Group | CIFAR-2 | Ensembles = 10
Projection dimension = 512
Training epoch = 20 |
| TRAK | Music Transformer Group | MAESTRO | Ensembles = 10
Projection dimension = 512
Training epoch = 10 |
| TRAK | GPT-2 Group | WikiText-2 | Ensembles = 5
Projection dimension = 2048
Training epoch = 3 |
| TRAK | 3-layer MLP | MNIST-10 | Ensembles $\in [1, 100]$ (specified in Section 4.3)
Projection dimension = 512
Training epoch = 50 |
| TRAK | Logistic Regression | MNIST-10 | Ensembles $\in [1, 100]$ (specified in Section 4.3)
Projection dimension = 512
Training epoch = 50 |
| influence function | All | All | Regularization = 1e-2
Training epoch = 50 |
| TracIn | All | All | Checkpoints = 1
Checkpoint select strategy: last one |
| rps | All | All | l2 regularization $\in \{1, 1e-1, 1e-2, 1e-3, 1e-4\}$
Selected as the value achieving the best LDS score within the range |

Table 5: Hyperparameter settings for different TDA methods across model and dataset configurations. The *Training epoch* denotes the number of epochs used for training the target model and is treated as an implicit hyperparameter, although it is not directly tied to the attribution method. For Influence Function, the *Regularization* refers to the damping term added when computing the inverse Hessian matrix $H^{-1}$. *All* means using the same hyperparameter settings across all model and dataset configurations.

## D   More Related Works

**Training Data Attribution.**   The objective of training data attribution is quantifying how each training sample affects the model's predictions by assigning an influence score. Most existing data attribution methods

can be categorized into two types (Hammoudeh & Lowd, 2024): retraining-based approaches (Cook & Weisberg, 1982; Ghorbani & Zou, 2019; Jia et al., 2019; Kwon & Zou, 2021; Wang & Jia, 2023; Ilyas et al., 2022) and gradient-based approaches (Koh & Liang, 2017; Yeh et al., 2018; Pruthi et al., 2020; Park et al., 2023; Kwon et al., 2023; Choe et al., 2024; Wang et al., 2024b; Bae et al., 2024). Retraining-based methods require training models on multiple subsets of the data, which leads to high computational costs, especially when applied to large-scale models or datasets. Gradient-based methods, in contrast, estimate data influence using the gradient signal of the trained model. These methods are generally more efficient and scalable, but many are based on assumptions such as model convexity and the validity of Taylor approximation. These assumptions are often violated in deep neural networks, leading to degraded performance and instability. Recent work (Søgaard et al., 2021; Park et al., 2023) shows that ensembling over independently trained models or checkpoints improves the gradient-based TDA's performance and robustness. This suggests that gradient-based methods still typically require internal access to the model and substantial computational resources for repeated training. This is also the key motivation of our work to explore gradient-based TDA under resource-constrained settings.

**TDA for Data Economics.** The data economy is a global digital ecosystem that creates economic value through the collection and exchange of data (Sestino et al., 2025). An essential problem that TDA aims to address in this domain is how to fairly allocate copyright or revenue across the data used for training, especially in the context of generative AI. Existing research has already made significant progress in leveraging data attribution methods for revenue and copyright compensation. For instance, Wang et al. (2024a) employs Shapley values for data valuation, while another study estimates the influence of training data on outputs in music generation models by using TRAK Deng et al. (2023), aiming to provide a quantitative measurement for enabling data providers to receive compensation proportional to their data impact. However, current studies are built on the assumption of full access to the target model. This raises a critical question: how can TDA remain feasible when model developers do not disclose or cannot share full model information? This serves as another motivation for our work.

## E    Table of Notation

| Notation | Description |
|---|---|
| $S = \{x_1, \ldots, x_n\}$ | Training dataset, where each $x_i = (x_i^{\text{feat}}, x_i^{\text{label}})$ |
| $\mathcal{S}$ | Space of all possible training datasets |
| $T = \{z_1, \ldots, z_m\}$ | Test dataset, where each $z_j = (z_j^{\text{feat}}, z_j^{\text{label}})$ |
| $\mathcal{T}$ | Space of all possible test datasets |
| $\phi = (\gamma, \eta)$ | Model configuration, consisting of architecture family $\gamma$ and hyperparameters $\eta$ |
| $\Phi$ | Space of model configurations |
| $\gamma$ | Model architecture family (e.g., Transformer, ResNet) |
| $\eta$ | Architecture-specific hyperparameters (e.g., number of layers, hidden size) |
| $\mathcal{A}$ | Training algorithm, $\mathcal{A} : \mathcal{S} \times \Phi \rightarrow \mathcal{F}$ |
| $f$ | Trained model obtained via $f = \mathcal{A}(S, \phi)$ |
| $\mathcal{F}$ | Set of all trained models |
| $Q(\cdot)$ | External query interface that returns the model prediction |

Table 6: Notations in Problem Definition

## F    TDA evaluation methods

**Linear Datamodeling Score (LDS).** For all evaluations conducted using LDS, we construct 50 randomly sampled subsets, each comprising 50% of the full training set. Accordingly, the parameter $\alpha$ in both Theorem 4.1 is set to 0.5.

**Area Under the ROC Curve for noisy label detection (AUC).** To evaluate how well a data attribution method can identify mislabeled training examples, we consider a setting where a portion of the training labels has been intentionally corrupted to simulate label noise. In our experiment, we randomly flip 10% of the whole training set. This creates a ground-truth split of the training data into clean and noisy subsets. The attribution method assigns a self-influence score to each training point, with the intuition that mislabeled (noisy) examples should generally receive higher scores than correctly labeled (clean) ones. The AUC is then computed as the probability that a randomly chosen noisy example is assigned a higher attribution score than a randomly chosen clean example. A higher AUC indicates better performance in distinguishing noisy labels from clean ones.

**Subset Removal Counterfactual Prediction (Brittleness Test).** This evaluation method assesses how the model's behavior changes when top-ranked training data points—identified by a data attribution method—are removed. Specifically, for classification tasks, we work on a number of (in our experiment, we use 100) test examples that are correctly classified when the model is trained on the full dataset. For each test point, we remove the top-k most positively influential training examples, retrain the model, and observe whether the prediction flips to an incorrect label. This counterfactual approach directly measures the quality of a TDA method by testing whether the identified influential examples are indeed crucial to the model's decision-making. As shown in fig. 5, all methods consistently outperform the random baseline, indicating that even untrained models can yield meaningful attribution scores in this task.

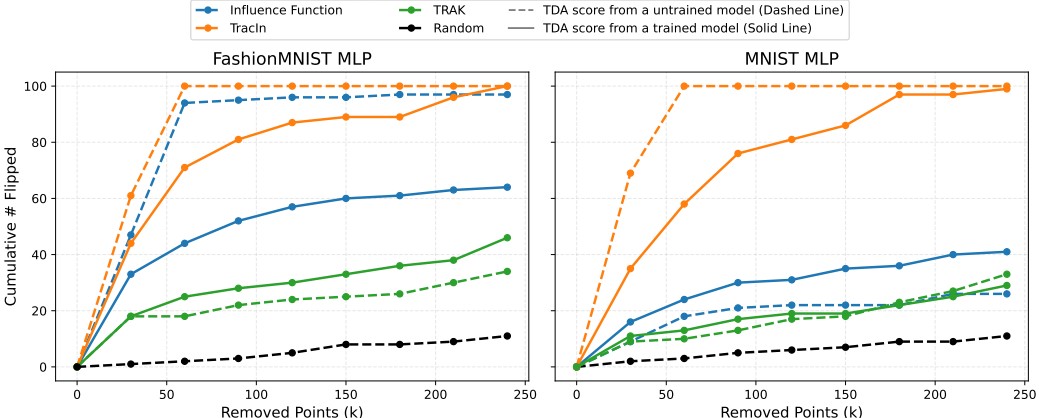

Figure 5: Brittleness Test results. As a supplement to the analysis in section 4.3, we also conducted the brittleness test on a 3-layer MLP model. In this experiment, we used two datasets: MNIST-10 (Deng, 2012) and FashionMNIST (Xiao et al., 2017). We first trained the MLP model on a subset of 500 training examples and selected 100 test examples that were correctly classified by the trained model. For each test point, we computed attribution scores using four different TDA methods under two settings: one with the fully trained model and the other with the random initialized model.

