# OpenReview forum: "Exploring Training Data Attribution under Limited Access Constraints"
_TMLR — Rejected by TMLR_

### Review · Reviewer_DhQk · 2025-11-14

**Summary Of Contributions:**

This paper investigated the feasibility of performing training data attribution (TDA) of a target model with different level of accessibility constraints using other proxy models, and claimed that: (1) it is feasible to perform TDA of a target model when partial model information is available (similar architecture); (2) infeasible to perform TDA when no knowledge is accessible; (3) calculating TDA scores on untrained models can yield non-trivial performance.

**Audience:**

No

**Audience Explanation:**

With all the confusions I had above, I do not recommend this paper to any TMLR audience.

**Claims And Evidence:**

No

**Claims Explanation:**

As an audience who is not very familiar with the topic (TDA), I'm very confused at reading this paper. Below are the main confusion points that I would like the authors to help me better understand the context and their research contribution.

## 1. The value to perform TDA on a limited access model

To my knowledge, TDA is a technique used to trace a model's prediction back to the specific training data points that were most influential and to influence down streaming tasks such as data curation for better training of such model. What is the purpose of TDA for a limited accessible model (which also means that we cannot perform training)?

## 2. Claim of considering the practical constraints that TDA methods may encounter in real-world scenarios.

Despite all listed constraints on the model accessibility (architecture, training strategy, weights, etc.), the most difficult part for me to understand is the assumption of full data access, which is mostly not true in real-world scenarios. To my knowledge, private companies view their data as much as, if not more than, their proprietary model artifacts. Therefore, "missing data access will largely (and more severely) break the data attribution task definition" is not sensible to me to justify this assumption. Neither is the citation of "ongoing legislative efforts to enhance the transparency of AI training data", because the requirement of disclosure of data sources is not equivalent to granting data access, thus is not supporting evidence that "missing data access" will not be the case for the future. Can the authors elaborate more on why the settings considered in this paper reflect real-world scenarios?

## 3. Claim of TDA using proxy model is feasible.

In my opinion, to show that "TDA using proxy model is feasible", the performance comparison should be set up with "the TDA performance using the original target model", and the results show negligible gap between these two. However, the experiments (e.g., Table 3) in this paper are set up to compare the "TDA performance of different proxy models", which is logically insufficient to support the claim.

On the other hand, the TDA performance in [1] (using the original target model) on setting "ResNet-9, CIFAR-10" is reported with an LDS score close to 0.6, whereas the best score in this paper on similar setting is 0.3338. Would it be more sensible conclude that "TDA using proxy model is bad"?

## 4. Trustworthiness of the experiment results

Will training the same proxy model using different training configs (E.g., different optimization algo., different scheduler, different random seed, etc) show consistent TDA performance? What is the statistical significance of the results (e.g., in Table 3)? Are this number derived from a single experiment?

[1] Park, Sung Min, et al. "Trak: Attributing model behavior at scale." arXiv preprint arXiv:2303.14186 (2023).

**Requested Changes:**

N/A

It would be much helpful if authors can help me clear my confusions

---

> ### Author Response · Authors · 2026-01-07
> **Official Comment by Authors**
>
> Thank you for your constructive feedback on our work. Below, we provide responses to your comments and questions.
>
> >[Question 1] The value of performing TDA under limited-access settings.
>
> Thank you for this question. As discussed in Section 1, training data attribution (TDA) supports a range of downstream applications, such as data selection and fact tracing. In particular, Appendix E discussed the role of TDA in data economy related scenarios, such as data revenue sharing and copyright allocation. Data providers and model developers are often not part of the same organization or company. We believe our study provides practical insights for such settings: even when data providers do not have access to the target model’s internal information, TDA can still be used to assess data contributions and offer quantitative evidence for fair compensation. Moreover, if TDA can be performed using untrained or small proxy models with close attribution quality, it enables low-cost data pre-selection and curation, as shown in Table 4.
>
> >[Question 2] Access to the training data set.
>
> Thank you for the thoughtful feedback. For TDA, as well as many downstream tasks that use TDA—such as data selection, data cleaning, and data compensation—full access to the training dataset is required. This is because these tasks rely on the relative ranking of a target data point (or a group of data points) within the complete training set, rather than the absolute influence value of an individual training example. The primary scope of this paper is to study the implementation of TDA under various access constraints. We acknowledge that the dependence on full data access remains a limitation of this work, and this limitation is explicitly discussed in Section 6 of the paper.
>
>
>
> >[Question 3] The feasibility of using proxy models.
>
> Thanks for your feedback.
> - We agree that the effectiveness of data attribution using proxy models should be evaluated against the TDA performance obtained by directly applying attribution to the original target model. Accordingly, as described in Section 4.2, we explicitly include a proxy model that is identical to the target model, serving as a baseline for the “full-access” or “perfect-guess” setting. Specifically, in each experimental group in Table 3, the boldfaced model on the left corresponds to attribution results computed directly using the target model, which served as a baseline.
> - We note that attribution quality, as measured by LDS, can vary across experimental settings and hyperparameter choices. As detailed in Appendix C, we specify the hyperparameters used for each setting in our experiment. In the ResNet-9 + CIFAR-2 experiments, we use an ensemble of 10 for TRAK. Under this configuration, the resulting LDS scores (0.3235) by using the target model ResNet-9 are close to those reported in prior work [1] for ResNet-9 + CIFAR-2 using 10 ensembles (see Figure E.1 Right), supporting the validity of our experimental results.
>
> [1] Park, Sung Min, et al. "Trak: Attributing model behavior at scale." arXiv preprint arXiv:2303.14186 (2023).
>
> >[Question 4] The reliability of the experimental results.
>
> Thanks for pointing that out. For TRAK, the ensemble strategy is used to improve robustness by averaging attribution scores computed from models trained with different random seeds. Accordingly, as described in Appendix C, across all three experimental settings, we compute TRAK scores using ensembles of proxy models trained with multiple random seeds to obtain more stable results.

---

> > ### Comment · Reviewer_DhQk · 2026-02-04
> > **Follow up comments**
> >
> > Thank you to the authors for the detailed explanation, which addresses most of my earlier questions.
> >
> > That said, I remain unconvinced by the following claim:
> >
> > “Accordingly, as described in Section 4.2, we explicitly include a proxy model that is identical to the target model, serving as a baseline for the ‘full‑access’ or ‘perfect‑guess’ setting.”
> >
> > The implication that such a proxy can serve as a substitute for the original model is problematic.
> >
> > Indeed, the manuscript states:
> >
> > >In particular, we include a proxy model that is architecturally identical to the target model, serving as a baseline for the ‘full‑access’ or ‘perfect‑guess’ setting.
> >
> > However, an architecturally identical model trained on the same data is not equivalent to the original model. Identical architecture and training data do not guarantee identical learned weights, and even small differences can lead to non‑trivial discrepancies in data attribution performance.

---

> > > ### Author Response · Authors · 2026-02-08
> > > **Follow up comments by authors**
> > >
> > > Thank you very much for the detailed feedback and questions!
> > >
> > > We would like to clarify that one of the popular data attribution methods that follows the derivation of the influence function, such as TRAK [1], LoGra [2], and EK-FAC [3], computes the data attribution score with respect to a training algorithm performed on a model with a training dataset. The randomness in the training process, which includes stochastic optimization, training order, and parameter initialization, is not fully explained by these methods.
> > >
> > > Another group of data attribution methods that accounts for a specific training trajectory, such as SGD-Influence[4] and DVEmb[5], computes the data attribution score with respect to a specific round of training. These methods often **require even stricter information**, such as per-example gradients at each training step, than the “Full Access” we proposed in the paper. Though they could calculate attribution scores that are “perfectly” aligned with a specific training trajectory, they are impractical for a use case with limited information access.
> > >
> > > [1] Park, Sung Min, et al. "Trak: Attributing model behavior at scale." PMLR, 2023.
> > >
> > > [2] Choe, S., et al. "What Is Your Data Worth to GPT? LLM-Scale Data Valuation with Influence Functions." NeurIPS, 2025.
> > >
> > > [3] Grosse, Roger, et al. "Studying large language model generalization with influence functions." arXiv preprint arXiv:2308.03296 (2023).
> > >
> > > [4] Hara, S., et al. "Data Cleansing for Models Trained with SGD." NeurIPS, 2019.
> > >
> > > [5] Wang, J. T., et al. "Capturing the Temporal Dependence of Training Data Influence." ICLR, 2025.

---

### Review · Reviewer_gJnt · 2025-12-22

**Summary Of Contributions:**

This paper studies training data attribution (TDA) for large black-box commercial models under realistic access and compute constraints. The authors formalize 5 constrained-access scenarios that capture common real-world limitations (lack of target model weights, inability to retrain variants, limited/no query access), while assuming the training data is available. They then evaluate proxy-based, gradient-style attribution (mostly via TRAK) across vision (CIFAR-2 / ResNet-9), music (MAESTRO / MusicTransformer), and language (WikiText-2 / GPT-2) tasks.

Empirically, they find: (1) proxy architecture match is crucial for useful attribution, (2) architecture-mismatched proxies remain poor even with query access and distillation, and (3) attribution computed on untrained or only-pretrained models can still be surprisingly informative—suggesting practical value for early-stage data curation. They also argue that the commonly used LDS metric does not directly assess attribution quality for a single checkpoint.

**Audience:**

Yes

**Audience Explanation:**

This should be of interest to TMLR's audience, specifically researchers working on interpretability, data valuation, dataset curation, and responsible ML, as well as practitioners interested in auditing model behavior when direct access to training internals is restricted. The paper’s “constrained access” framing closely matches real-world settings for commercial models.

**Claims And Evidence:**

Yes

**Claims Explanation:**

The main claims are backed by targeted experiments that vary the exact factors the paper highlights: proxy architecture match/mismatch, different access scenarios (including limited/no query), and training state of the proxy (random init vs pretrained vs finetuned). The results are shown consistently across multiple modalities and tasks, which suggests that the observed trends are not domain-specific. The discussion around limitations of LDS also appears well-motivated given the paper’s framing around evaluating attribution for a fixed checkpoint rather than across training trajectories.

The evidence is most convincing within the scope tested (small-to-mid-scale models, TRAK-centric pipeline). The paper would be stronger with a clearer argument about how these findings should extrapolate to modern frontier-scale LLMs (very deep transformers, possible mixture-of-experts, different training dynamics), and with a bit more diversity in attribution approaches.

**Requested Changes:**

1. Clarify the surprising GPT-2 result -- would strengthen
The paper reports that TRAK on pretrained GPT-2 can yield LDS / data-selection performance slightly better than TRAK on the fine-tuned model. This is surprising. Does fine-tuning overfit the representation for the downstream task in a way that hurts global attribution? Is the signal dominated by general-purpose features learned in pretraining? Are there optimization artifacts (e.g., feature collapse, reduced gradient diversity) that affect TRAK embeddings?

2. Discuss expected scaling to frontier LLMs and MoE -- critical
Since the motivation emphasizes commercial frontier models, including a more explicit discussion of how findings might translate (or fail to translate) to much larger transformers and MoE architectures would strengthen the paper. If experiments at scale are out of scope, a careful limitations section is still valuable: what aspects should remain invariant (e.g., importance of architecture match), and what might change (e.g., distillation efficacy, proxy capacity requirements, representation transfer)?

3. Broaden the methodological picture beyond TRAK -- would strengthen
The study uses TRAK and related influence/TracIn methods. Including one different attribution family or proxy strategy (even at small scale) or would be useful.

---

> ### Author Response · Authors · 2026-01-07
> **Official Comment by Authors**
>
> Thank you for recognizing the motivation of our work and the empirical evidence supporting our main claims. We respond to the comments below.
>
> >[RC 1]  TDA results for pre-trained models.
>
> Thank you for raising this insightful question. We agree that it is surprising that TRAK on pretrained GPT-2 can slightly outperform the fine-tuned model in LDS performance, and we view this as a practical property that lowers the computational cost for TDA. Understanding the underlying mechanisms is an important direction for future work.
>
> >[RC 2] Scalability to large commercial models.
>
> Thank you very much for your feedback. In our experiments, we include a relatively large experimental setting. Specifically, for both the proxy-model experiments and the experiments using pretrained models, we adopt a GPT2 architecture with up to 124 M parameters and train it on the WikiText dataset. However, due to the computational cost and practical limitations of existing data attribution methods, scaling such experiments to frontier, production-level commercial models remains challenging. However, we believe our findings provide valuable insights and can inform future research in this direction. We will explicitly discuss these limitations and their implications in our final draft.
>
> >[RC 3] Extending to other categories of attribution methods.
>
> Thank you very much for your suggestion. We explored incorporating additional classes of attribution methods, including Representer Point Selection (RPS), into the proxy-model setting. However, RPS suffers from relatively low attribution quality in practice and does not readily scale to larger experimental settings, such as Music Transformer or GPT-style models, as reported in [1]. Since this class of methods already exhibits limited attribution performance even when applied directly to the target model, extending them to proxy-based attribution remains constrained, even when the proxy model closely approximates the target. Regarding retraining methods, such as Datamodels[2], they are limited to the relatively high computational cost.  Since this paper focuses on relatively practical implementations, these methods were not included in the discussion.
>
> [1] Deng, Junwei, et al. "dattri: A Library for Efficient Data Attribution."  arXiv preprint arXiv:2410.04555 (2024).
>
> [2] Ilyas, Andrew, et al. "Datamodels: Understanding predictions with data and data with predictions." In International Conference on Machine Learning (ICML), 2022.

---

### Review · Reviewer_naST · 2025-12-23

**Summary Of Contributions:**

This work is an empirical investigation of how existing training data attribution methods behave under increasingly constrained access to the target model and its training process. It focuses on proxy models, distillation, and even untrained or pretrained-only networks. Its main contribution is some empirical observations about when attribution scores remain usable and when they degrade. It does not introduce new methods, theory, or evaluation principles (I am not saying that it should). A strength of the paper is the systematic nature of the stress tests across several constraint regimes and attribution techniques, including some negative observations (e.g., that distillation does not reliably improve attribution fidelity). However, the contribution is limited in scope: several conceptual aspects overlap with prior work on final-model-only and distributional attribution. The paper does not provide a deeper explanation or formalization of the observed phenomena. Moreover the experimental evidence is based on relatively small datasets and models (including MNIST), which raises questions about the robustness and generality of the conclusions, especially for the large-scale settings that motivate the paper.

**Audience:**

No

**Audience Explanation:**

My answer to the previous question is yes -- in the sense that many people are interested in this area/problem. But I am not convinced that this paper would help this community move forward, based on solid conclusions,.

I think it is important that we examine the contribution of this paper relative to what is already known in the recent literature, still satisfying the TMLR editorial policies.

So, relative to the current state of the art in training data attribution, this paper can be viewed as a complementary empirical study rather than a new or unifying reference. Recent work such as Final-Model-Only Data Attribution (Wei et al., 2024) provides a more clear and more principled framing of access-constrained attribution settings. Distributional Training Data Attribution (Mlodozeniec et al., 2025) offers a deeper conceptual understanding of what influence-based methods measure in the presence of training stochasticity and reinterprets common evaluation metrics such as LDS. Surrogate-based approaches such as Wrapper Boxes (Su et al., 2024) address limited-access settings through model redesign with explicit faithfulness guarantees. Compared to these works, the present paper does not advance the methodological or theoretical frontier but instead reports empirical observations about the behavior of existing attribution methods under a collection of constrained scenarios. Consequently, it is not clear to me that the paper would serve as a better or more authoritative resource for readers interested in this research problem than these recent contributions.

**Broader Impact Concerns:**

I do not see any such concerns.

**Claims And Evidence:**

No

**Claims Explanation:**

While the empirical results are presented well, several of the paper’s claims appear broader than what the evidence can reasonably support given the limited scale and scope of the experiments.

First the paper suggests that training data attribution can remain “meaningful” even when computed using randomly initialized or merely pretrained models, based on results for noisy-label detection and data selection on a small number of benchmark datasets. I am skeptical that these observations would generalize to larger, more heterogeneous datasets, more complex label structures, or tasks where the learned representations differ substantially from random or generic features. In particular, it is unclear whether such signals would persist in settings with  higher input dimensionality, stronger class imbalance, or more subtle spurious correlations, where attribution methods are known to be fragile

Second, the conclusion that architectural family matching matters far more than model size for proxy-based attribution is drawn from a relatively narrow set of architectures and scales. This observation may not hold in regimes where scaling qualitatively changes the representation geometry (eg. very large pre-trained models or models trained with different objectives). I am also concerned about regimes in which model size interacts with optimization dynamics and implicit regularization in ways that are not captured by the experiments.

Third, the claim that distillation improves output matching but does not meaningfully improve attribution alignment is supported by a small number of distilled student–teacher pairs. This conclusion could depend strongly on the specific distillation procedure, temperature, loss design, or training duration. It is not evident to me that this negative result would persist under more carefully optimized or task-specific distillation strategies

Finally the paper uses attribution quality metrics such as LDS to draw general conclusions about what attribution methods “measure.” However, these metrics are known to be sensitive to stochasticity, dataset size, and evaluation protocol. Without broader validation across larger datasets, more diverse tasks, and multiple random seeds, it is difficult to justify the paper’s more general claims about the reliability and interpretation of attribution under constrained access.

**Requested Changes:**

The paper says that attribution scores computed using randomly initialized or merely pretrained models can be useful for noisy-label detection and data selection. Can you clarify what properties of these models are essential for this effect? For example, have you tested whether the signal persists across different random initializations, different pretrained checkpoints, or different pretraining objectives, and how sensitive the results are to these choices?


The paper concludes that distillation improves output matching but does not improve attribution alignment. Plz provide more detail on the distillation procedures used (e.g., temperature schedules, loss weighting, training duration) How confident are you that the observed negative result is not an artifact of a suboptimal or limited distillation setup?

Given that attribution methods and evaluation metrics are known to be sensitive to training randomness, can you plz clarify how many random seeds were used throughout the experiments? Are the reported trends stable across seeds, especially in the more surprising regimes involving untrained or randomly initialized models?

The paper uses LDS-based evaluations to draw conclusions about what attribution methods “measure” under limited access. How do you reconcile these interpretations with recent work that explicitly frames attribution as a distributional quantity over training randomness? Would the conclusions change if a distributional variant of LDS were used?


Most of the conclusions are based on experiments with TRAK and closely related methods. Do the authors believe these findings generalize to other classes of attribution methods (e.g., representation-based or non-gradient-based approaches), and have you tested any such alternatives?

---

> ### Author Response · Authors · 2026-01-07
> **Official Comment by Authors**
>
> Thank you for acknowledging our experimental presentation. We reply to the comments as follows:
>
> > [RC 1] Regarding the randomly initialized models.
>
> Thank you for this question. Across all three experimental settings (MNIST-10 + LR, MNIST-10 + MLP, and CIFAR-2 + ResNet-9), we consistently observed that randomly initialized models can yield some informative attribution results. As an additional analysis, we evaluate LDS scores under different random model initialization methods in the CIFAR-2 + ResNet-9 setting, following the same configuration as Section 4.3.1 with 5,000 training samples and 500 test samples.
>
> | Init Method | TRAK-1 | TRAK-10 | TRAK-20 | TRAK-50 |
> |------------|--------|---------|---------|---------|
> | He         | 0.0887 | 0.1991  | 0.2265  | 0.2511  |
> | Xavier    | 0.0996 | 0.2196  | 0.2489  | 0.2685  |
> | LeCun     | 0.1050 | 0.2224  | 0.2428  | 0.2682  |
>
>
> While there are variations in results across different initialization methods, the results are broadly similar and exhibit the same increasing trend as the TRAK ensemble number grows. We believe it can motivate future work to better understand the key properties underlying this behavior.
>
> > [RC 2] Regarding the distillation results.
>
> Thank you for pointing out the potential sensitivity issue of the distillation experiment.  We provide detailed hyperparameter settings for the distillation training in Appendix A.2. And as discussed in Section 4.2, we only evaluate those proxy models whose output distributions on the test set are more similar to that of the target model—specifically, those that achieve lower KL divergence than their corresponding non-distilled counterparts. We believe that the lack of improvement in data attribution quality stems from the following reason: under limited-access settings, distillation aligns the proxy model with the target model solely through output-level signals. Discrepancies in architecture and parameterization between distilled and target models result in divergent gradient structures. As a result, gradient-based data attribution methods may struggle to approximate the attribution behavior of the target model.
>
> >[RC 3] Regarding the randomness of training.
>
> We specify the detailed hyperparameter configurations for our experiments in Appendix C. We consistently employ ensemble-based strategies for TRAK. For example, in the ResNet experiments, all TRAK scores are obtained by ensembling TDA scores computed from models trained with 10 different random seeds. In the random initialization experiments, the ensemble size for TRAK is further increased to 100 to ensure robustness. For LDS, the ground-truth values are also obtained by averaging 50 randomly sampled subsets with a 50% sampling ratio, to ensure robustness.
>
> > [RC 4] Regarding distributional attribution.
>
> We thank the reviewer for highlighting the work on distributional training data attribution [1]. The challenges posed by stochasticity (e.g., initialization and batching) are inherent to training data attribution, regardless of the access constraints discussed in our paper. While the concept of distributional attribution is orthogonal to our current scope, its integration represents a promising direction.
>
> > [RC 5] Extending to other categories of attribution methods.
>
> Thank you very much for your suggestion. We explored incorporating additional classes of attribution methods, including Representer Point Selection (RPS), into the proxy-model setting. However, RPS suffers from relatively low attribution quality in practice and does not readily scale to larger experimental settings, such as Music Transformer or GPT-style models, as reported in [2]. Since this class of methods already exhibits limited attribution performance even when applied directly to the target model, extending them to proxy-based attribution remains constrained, even when the proxy model closely approximates the target. Regarding retraining methods, such as Datamodels[3], they are limited to the relatively high computational cost.  Since this paper focuses on relatively practical implementations, these methods were not included in the discussion.
>
> [1] Mlodozeniec, B., Reid, I., Power, S., Krueger, D., Erdogdu, M., Turner, R. E., & Grosse, R. (2025). Distributional Training Data Attribution. arXiv preprint arXiv:2506.12965.
>
> [2] Deng, Junwei, et al. "dattri: A Library for Efficient Data Attribution."  arXiv preprint arXiv:2410.04555 (2024).
>
> [3] Ilyas, Andrew, et al. "Datamodels: Understanding predictions with data and data with predictions." In International Conference on Machine Learning (ICML), 2022.

---

### Decision · Action_Editor_WmPA · 2026-04-09

**Recommendation:** Reject

**Audience:**

Yes

**Audience Explanation:**

The reviewers were split on this question, but generally thought the answer should be yes. I will note, however, that Reviewer naST was unsure whether this paper offers new insights beyond existing work on TDA, so it would be useful to further address this point in a major revision.

**Claims And Evidence:**

No

**Claims Explanation:**

This paper studies existing training data attribution (TDA) methods under constrained access to the target model and its training process. The reviewers largely agreed that the problem is timely and practically motivated. At the same time, the reviewers disagreed on whether the claims are sufficiently supported by clear and convincing evidence, primarily because the experiments were viewed to be too limited in scale/coverage for some of the paper’s broader claims. I thank the reviewers for their constructive reviews and the authors for their thoughtful responses!

I think that many of the reviewers’ concerns could be allayed by narrowing the framing/scope of the claims, clarifying the proxy-model baseline discussion, and strengthening the empirical support (e.g., by running some larger-scale experiments), so I recommend a “major revision” (i.e., revise and resubmit).

**Resubmission Of Major Revision:**

The authors may consider submitting a major revision at a later time.